



# Monitoring Tropical Debris Covered Glacier Dynamics from High Resolution Unmanned Aerial Vehicle Photogrammetry, Cordillera Blanca, Peru

Oliver Wigmore[1,2,3,4], Bryan Mark[1,2]

[1]Department of Geography, The Ohio State University, Columbus OH, USA
[2]Byrd Polar and Climate Research Center, The Ohio State University, Columbus OH, USA
[3]Institute of Arctic and Alpine Research, University of Colorado Boulder
[4]Earth Lab, University of Colorado Boulder

*Correspondence to*: Oliver Wigmore (oliver.wigmore@colorado.edu)

**Abstract.** The glaciers of the Cordillera Blanca Peru are rapidly retreating as a result of climate change, altering timing, quantity and quality of water available to downstream users. Furthermore, increases in the number and size of proglacial lakes associated with these melting glaciers is increasing potential exposure to glacier lake outburst floods (GLOFs). Understanding how these glaciers are changing and their connection to proglacial lake systems is thus of critical importance. Most satellite data are too coarse for studying small mountain glaciers and are often affected by cloud cover, while traditional airborne photogrammetry and LiDAR are costly. Recent developments have made Unmanned Aerial Vehicles (UAVs) a viable and potentially transformative method for studying glacier change at high spatial resolution, on demand and at relatively low cost.

Using a custom designed hexacopter built for high altitude (4000-6000masl) operation we completed repeat aerial surveys (2014 and 2015) of the debris covered Llaca glacier tongue and proglacial lake system. High resolution orthomosaics (5cm) and digital elevation models (DEMs) (10cm) were produced and their accuracy assessed. Analysis of these datasets reveals highly heterogeneous patterns of glacier change. The most rapid areas of ice loss were associated with exposed ice cliffs and melt water ponds on the glacier surface. Considerable subsidence and low surface velocities were also measured on the sediments within the pro-glacial lake, indicating the presence of extensive regions of buried ice and continued connection to the glacier tongue. Only limited horizontal retreat of the glacier tongue was observed, indicating that simple measurements of changes in aerial extent are inadequate for understanding actual changes in glacier ice quantity.

## 1 Introduction

The Peruvian Andes hold around 70% of the world's tropical ice. Within Peru the Cordillera Blanca is the most glacierised mountain range. These glaciers are rapidly retreating primarily as a result of warming temperatures (Vuille et al., 2008; Vuille et al., 2003). Most recently Burns and Nolin (2014) calculated a 25% reduction in glacier area from 1987 to 2010. Similarly, Racoviteanu et al., (2008) calculated a 22.4% reduction from 1970 to 2003 with an increase in the rate of retreat



from 1970, and a rise in glacier terminus elevation of 113m. Glacier changes are altering downstream hydrology, changing water supply in terms of quantity, quality and timing (Baraer et al., 2009, 2012; Chevallier et al., 2011; Condom et al., 2011; Juen et al., 2007; Kaser et al., 2003; Rabatel et al., 2013a). These changes have profound impacts on downstream communities, livelihoods and ecology (Bury et al., 2013, 2011; Carey et al., 2014; Mark et al., 2010; Postigo et al., 2008; Young, 2009,
2014). Additionally, rapidly melting glaciers are increasing the number and volume of unstable proglacial lakes which represent a serious natural hazard through potential glacier lake outburst floods (GLOFs) (Carey, 2008; Carey et al., 2012; Huggel et al., 2002; Lliboutry et al., 1977; Portocarrero, 2014a). Processes of glacier melt and the formation of glacier lakes is controlled by a multitude of factors including local energy balance, topography, geology, aspect, valley dimensions, moraine stability, debris thickness and the presence of surface features such as melt ponds and ice cliffs (Benn et al., 2012; Buri et al.,
2016; Harrison et al., 2006; Huggel et al., 2002; Immerzeel et al., 2014; Lejeune et al., 2013; Portocarrero, 2014a; Reid et al., 2012; Reid and Brock, 2010; Thompson et al., 2012). The impact of these variables has been studied extensively in other areas, particularly the Himalaya, where large debris covered glacier tongues are common (Benn et al., 2012; Buri et al., 2016; Immerzeel et al., 2014; Scherler et al., 2011; Shroder et al., 2000). However, only one study has investigated geomorphic change for a debris covered glacier in the Cordillera Blanca (Emmer et al., 2015). Understanding how these glaciers are
changing, their connection to proglacial lake systems and the role of glacier surface heterogeneity in controlling melt processes is critical to inform future management and adaptation strategies. Unfortunately, current understanding has been limited due to the spatial and temporal resolution of available datasets and difficulties of accessibility to these remote mountain locations.

## 1.1 Quantifying Glacier Change

There are four primary methods for quantifying glacier change: glaciological measurement; energy balance and degree day
modelling; hydrologic reconstruction; and, digital elevation model (DEM)/remote sensing analysis. Glaciological methods include all types of direct measurement such as installation of ablation stakes on the glacier surface, topographic survey and GNSS data collection. These methods collect precise data at a given point which are then extrapolated to the glacier as a whole (Francou et al., 2003; Hastenrath and Ames, 1995a; Kaser et al., 1990; Ribstein et al., 1995). For glaciers with uniform topography and albedo properties this method is usually acceptable, however when the glacier surface is highly heterogeneous
(e.g. debris cover, melt ponds, cliffs etc) such as those in the Cordillera Blanca this approach can be inaccurate. Furthermore, access to many of the glaciers in the Cordillera Blanca is made difficult though extensive crevasses, avalanche paths and thick debris cover.

Energy balance methods estimate glacier volume changes through the calculation of melt water generation as a function of
controlling climatic and surface variables, e.g. incoming solar radiation, ambient air temperature, relative humidity, albedo, debris thickness, etc (Hock, 2005; Hock and Holmgren, 2005; Lejeune et al., 2013; Reid et al., 2012; Sicart et al., 2011; Wagnon et al., 1999). Numerous studies have investigated climatic controls on glacier mass balance within the Cordillera Blanca  and the tropical Andes (Francou et al., 2003; Hastenrath and Ames, 1995a; Mark and Seltzer, 2005; Rabatel et al.,



2013b; Ribstein et al., 1995; Sicart et al., 2011). Additionally, energy balance models have been used to investigate the role of debris cover on glacier melt (Lejeune et al., 2013; Reid et al., 2012; Reid and Brock, 2010) and evolution of surface features (Buri et al., 2016), though not in the Cordillera Blanca. Energy balance models can provide unique insights into processes that drive glacier melt but are often limited in quantifying precise melt rates due to the heterogeneity of input variables over the

glacier surface (La Frenierre and Mark, 2013; Hock, 2005; Hock and Holmgren, 2005; Reid et al., 2012). Degree day models simplify this approach by relying solely on empirical relationships between ablation and air temperature. However, degree day models typically do not work well for tropical glaciers as thermal variation is minor and poorly correlated with glacier melt (Juen et al., 2007). Elsewhere, these models can provide good mass balance estimates at the basin scale but provide limited information on surface melt processes (Hock, 2003, 2005; Pellicciotti et al., 2005).

Hydrologic reconstruction essentially uses measured discharge and precipitation records to reconstruct glacier mass balance changes through hydrologic modelling and partitioning of the different components. This method has found some success in the Cordillera Blanca, where direct measurements and meteorological datasets are sparse (Baraer et al., 2012; Kaser et al., 2003; Mark and Seltzer, 2003; Vuille et al., 2008). However hydrological reconstruction of glacier mass balance can only be

applied at the glacier/catchment scale and provides no information on glacier surface dynamics.

Remote sensing methods utilise terrestrial, airborne and satellite remotely sensed data including multispectral imagery, RADAR, LiDAR, etc to create orthomosaics and digital elevation models (DEMs). Through differencing across multiple dates these datasets can be used to understand changes in glacier volume, surface velocities and melt patterns (Bamber and Rivera,

2007; Hastenrath and Ames, 1995b; Heid and Kääb, 2012; Immerzeel et al., 2014; Kraaijenbrink et al., 2016; Mark and Seltzer, 2005; Nuth and Kääb, 2011; Peduzzi et al., 2010; Racoviteanu et al., 2007, 2008). These methods provide a spatially continuous means of interpreting glacier change, and are ideal in locations where accessibility is difficult. However, they are limited by the spatial and temporal resolution and vertical accuracy of the available data (Nuth and Kääb, 2011; Racoviteanu et al., 2007). Widely available elevation products e.g. ASTER and SRTM have decimetre resolutions and poor vertical accuracy (for

detecting minor elevation changes), while higher resolution products e.g. aerial photogrammetry, LiDAR and RADAR are costly to acquire. As a result DEM differencing has not seen widespread use in the Cordillera Blanca due to the extreme topographic variation and generally small glacier size (typically less than $0.5km^2$). Mark and Seltzer (2005) integrated DEM differencing and climate variables to investigate the role of aspect and elevation on glacier ablation. Huh (2014) and Huh et al., (2012) combined airborne LiDAR and photogrammetry to quantify changes in glacier volume for a handful of valleys within the Cordillera Blanca, but large errors exist for the earlier date photogrammetric DEM's. Most recently Emmer et al.,

(2015) combined a single DEM with high resolution satellite imagery and field investigations to understand the geomorphic evolution and measure surface velocities of the debris covered Jatunraju glacier. An additional remote sensing based method applies a scaling factor to convert glacier are measurements to volume (Bahr et al., 1997). However, this method is often problematic, especially for tropical glacier where variable hypsometries, extreme topographic variations and variable glacier



sizes can affect the accuracy of derived glacier volumes and thus volume changes (Baraer et al., 2012; Salzmann et al., 2013). Furthermore, this method can only provide information for the entire glacier, i.e. there is no information on spatial variability.

## 1.2 Unmanned Aerial Vehicles

Traditional methods of remote sensing (satellite and airborne) are constrained by three competing primary factors: spatial resolution, temporal resolution and cost. To make a gain in one of these factors there is a simultaneous loss in one or both of the other factors. Additionally, in mountain regions cloud cover is a notable issue as is safety for airborne platforms. Recent developments in military, remote control, hobby and open source hardware have made it viable and potentially transformative to consider new small (1-5 kg), inexpensive UAV platforms for mapping glacier dynamics. UAV platforms include, helicopters, multi-rotors, fixed wings and paragliders; they are typically small (1kg - 5kg) and relatively cheap, ranging in cost from $400 to $40,000 (Eisenbeiß et al., 2009; Eisenbeiss and Sauerbier, 2011; Friedli, 2013). These platforms may be operated by remote control, first person or GPS enabled autopilot systems and can be fitted with a diversity of sensors for collecting observations of the earth surface and atmosphere, including multispectral cameras, meteorology sensors and LiDAR units (Colomina and Molina, 2014; Hardin and Jensen, 2011; Nex and Remondino, 2014). Due to the low flight elevations (50 - 1000m above ground level) very high resolution imagery (<5cm pixels) can be collected from below the cloud level with limited risk to ground personnel or property. In addition these on-demand platforms can collect data at whatever temporal resolution is desired with no major additional acquisition costs.

In addition to these developments in UAV hardware, software developments have also occurred. Easy to use off the shelf software is now capable of processing large UAV image collections to derive orthomosaics and dense photogrammetric point clouds for the generation of digital elevation models (DEMs). By integrating these data with differential GPS surveyed ground control points (GCP's) it is possible to generate DEMs with a ground resolution of under 5cm and orthomosaics of around 2cm resolution (Harwin and Lucieer, 2012; Hugenholtz et al., 2013). Accuracy of these datasets has been demonstrated to be very high in comparison to survey data and often equal or better than LiDAR (Harwin and Lucieer, 2012; Hugenholtz et al., 2013).

The use of UAVs in earth science research is undergoing rapid expansion with numerous researchers exploiting the financial and spatiotemporal resolution benefits of UAV technology (Ahmad, 2011; Colomina and Molina, 2014; D'Oleire-Oltmanns et al., 2012; Hardin and Jensen, 2011; Koh and Wich, 2012; Vivoni et al., 2014; Watts et al., 2012). A few projects have utilised UAV platforms in mountainous glaciated regions (Bhardwaj et al., 2016). Friedli (2013) has investigated their potential for glacier monitoring on the Rhone glacier, while Whitehead et al., 2013 conducted a detailed glacier study in the Canadian Arctic, however this was at low elevation. At higher elevations M. Willis (pers. com.) and S. Wernke (pers. com.) have had success using multirotor and fixed wing platforms for archaeological mapping in the Ecuadorian and Peruvian Andes respectively, at altitudes of up to 4000m, and a videographer has used a multi-rotor system to film climbers on the Trango Towers at 6000m in Pakistan (https://vimeo.com/50029357). Recently Immerzeel et al., (2014) and Kraaijenbrink et al., (2016)



successfully used a small fixed wing platform to map glacier changes and surface velocities at 5000m in the Himalaya. However, to our knowledge no studies to date have deployed UAVs to study glacier dynamics within the Cordillera Blanca or the Tropical Andes.

## 2 Objectives

There are four principle objectives of this study:

1) Successfully deploy a multirotor platform suitable for mapping glacier changes and capable of operating at ~5000masl in the mountains.

2) Measure glacier volume changes, surface velocities and proglacial lake changes at high resolution using decimetre DEM's and centimetre orthomosaics.

3) Use these data sets to investigate processes and patterns of glacier change over a debris covered glacier tongue in the Cordillera Blanca.

4) Investigate connectivity between the glacier and proglacial lake system and potential future evolution of this reservoir.

## 3 Study Area

The Cordillera Blanca is located in the central Andes of Peru (Fig. 1). The range extends 180km (north south) encompassing
numerous peaks over 6000m, including Peru's highest, Huascaran at 6768m. The region contains more than 700 individual glaciers which terminate at over 4500masl. The climate is dominated by a strong wet/dry seasonal signal. The wet season corresponds to the austral summer (October-May) with roughly 80% of the 800-1200mm/yr of precipitation (Bury et al., 2011; Mark et al., 2010) falling over this time. As with other tropical mountain regions diurnal temperature variation greatly exceeds seasonal temperature variability (Kaser, 2001). Maximum ablation occurs at the same time as maximum accumulation during
the wet season months (Kaser, 2001). The focus of this study is Llaca glacier, which is located in the central Cordillera Blanca above the city of Huaraz (Fig. 1).

The Llaca valley is relatively narrow constricting to around 500m at its narrowest. The head of the valley opens into a large cirque over 3km wide with headwalls over 5500m and includes the summits of Ranrapalca (6162m) and Vallunaraju (5686m).
Due to this large accumulation zone and narrow valley Llaca glacier extends to a considerably lower elevation (4500masl) than many other glaciers in the region which typically terminate between 4800-5100masl. The glacier snout is covered in a thick (>1m) layer of debris derived from the collapse of steep lateral moraines onto the glacier surface, which insulates the ice and also enables its extension to this lower terminal elevation. The lower debris covered tongue remains connected to the upper glacier and terminates in a moraine dammed and partially ice cored proglacial lake approximately 1km long, 0.2km wide and
17m deep (in 2004) storing an estimated 274300m$^3$ of liquid water (Portocarrero, 2014b). The moraine dam underwent



stabilisation and lowering (completed in 1977) as a component of the broader campaign within the Cordillera Blanca to mitigate the risk of glacier lake outburst floods (GLOF's) which have been responsible for a number of major natural disasters and large loss of life in the region (Carey, 2008; Carey et al., 2012; Lliboutry et al., 1977; Portocarrero, 2014b). Lake stabilisation included construction of a reinforced concrete and steel dam within the terminal moraine and artificial lowering

of the lake level. Despite these efforts a significant GLOF risk is still present due to avalanche risk from hanging glaciers above, tectonic activity and the location of this lake directly above the city of Huaraz (Portocarrero, 2014b).

## 4 Methods

### 4.1 UAS Platform Design and Specifications

A hexa-multirotor UAV was custom designed and built for this project (Fig. 2). The UAV is capable of operating at elevations

between 3500 and 6000m above sea level. The 1m (diameter) frame utilises a lightweight carbon fibre construction and is fitted with high speed motors and 13 to 15 inch carbon fibre propellers (depending on flight altitude). All up weight excluding cameras is ~2kgs. Low weight is essential for operation at this altitude and eases transport when hiking/climbing to the glaciers. The maximum flight time for the platform at 5000masl is around 12-15 minutes on a single 10,000MAH 4S Lithium Polymer (LiPo) battery giving a range of 1.5-2.5km (depending on above ground flight altitude) from launch at flight speeds acceptable

for aerial mapping (~8m/s). Autonomous navigation is managed by a 3DR Pixhawk flight controller. In comparison to fixed wing platforms; a multirotor platform sacrifices flight time in exchange for greater navigational precision, vertical takeoff and landing (essential over debris covered glacier surfaces and around water bodies), and typically higher quality images due to greater platform stability. Communication with the UAV is maintained with an extended range (2.5km) 2.4GHz radio controller (FrSky Taranis LR) and a long range (>40km) ~915MHz (RFD900+) telemetry link between the autopilot and

ground station. Ground station control was managed by a field tablet running APM Mission Planner for Android (2014) and APM Mission Planner for Windows (2015).

The UAV was fitted with a single Canon S110 Powershot camera. The camera was installed in nadir view with no gimbal stabilisation. This camera allows full manual control of exposure settings including; ISO, F-stop, and shutter speed. The best

settings are an ISO ~100-200 (to reduce image graininess), F-stop >4 (to improve depth of field), and shutter speed <1/1000s (to reduce motion blur). Camera control was managed by the Canon Hack Development Kit (CHDK) (chdk.wiki.com). CHDK reboots the camera using firmware stored on the SD card and allows the use of custom scripts. In this case the kap_uav.lua script (chdk.wikia.com/wiki/KAP_UAV_Exposure_Control_Script) was used. This script allows the specification of the camera parameters above and automatic image capture at set intervals. An interval of three seconds was selected to provide

>90% image overlap, with additional redundancy for blurred image removal.



### 4.2 Ground Control

Before each survey a comprehensive GNSS survey was conducted. This included installation and survey of highly visible targets for use as ground control points (GCP's) in georectification of the SfM point cloud and collection of 'check points' for accuracy assessment of derived DSM's (Fig. 2, Fig. 3, Table 1). Target installation points were distributed fairly evenly across the survey area making sure to include high and low points within the survey region. Access to the upper portion of the study area is very difficult and targets were unable to be installed here. Targets were anchored to the ground using metal stakes and rocks as appropriate to ensure they would not move before flights. Selection of check points was semi-random making sure to include high and low points within the land surface. In 2014 points were taken atop and between individual rocks as well as over more uniform terrain. In 2015 check points were placed primarily in areas that exhibit relative terrain homogeneity within a roughly 50cm radius.

In 2014 base station data (for rover position correction) was collected from an apartment roof in Huaraz (baseline ~14km) and occupied for almost 7 days (Table 2). In 2015 base station data was collected at the southern end of the moraine in Llaca valley (baseline ~1km) and occupied for 6.5 hours. Base station positions were resolved using the Natural Resources Canada (NRCAN) Precise Point Positioning online service. Rover positions (targets and check points) were collected using a 5 minute occupation fast static methodology (Table 2). Following data collection all points were processed using Topcon Magnet™ Office Tools (version 2.7.1) software with base station coordinates adjusted as per NRCAN positioning results.

### 4.3 Aerial Survey Flight Paths

Flight planning was completed in the office using mission planner software and Google Earth. As the UAV is does not have collision avoidance it is critical to carefully plan a flight path that avoids topographic obstacles such as moraine walls, cliffs etc, especially considering that for much of the survey the UAV would be flying beyond line of sight. Much of Google Earths topographic data relies on SRTM90 which is both coarse resolution and often dated for use in the dynamic and heterogeneous terrain of the Cordillera Blanca. Additionally, limited sky view and possible dropped satellites present real navigational threats for the UAV itself. Therefore an above ground level (agl) of ~100m was selected; as this would provide sufficient ground resolution (~3cms) while also keeping the UAV well above the moraine walls (Fig. 3, Table 3). To maintain this above ground altitude the flight plan required the UAV to first ascend to 100m, then gain an additional ~200m as it moved up the valley over the glacier. This also maintains a uniform ground resolution. Five flight lines were constructed from the launch point below the calving face, to provide ~60% sidelap which is necessary for robust DEM recovery (Fig. 3). The maximum length of an individual flight line was 1.2km from the take off point, resulting in a total (return) flight length of roughly 2.5km. Total flight time was around 12mins per flight travelling at ~8m/s. Rapid battery drain was experienced in the takeoff portion and when gaining altitude up valley which limited the range and thus the total survey-able region.



## 4.4 Structure from Motion Workflow - DEM and Orthomosaic Generation

Collected image data were processed using the Structure from Motion workflow as implemented in Agisoft PhotoScan Professional Edition (Version 1.1) (Agisoft, 2016). This is proprietary software thus the specific algorithms used are not available. However, the basic principles outlined below are similar across all commercial and open source software packages. Further details on SfM can be found in Fonstad et al., (2013), Lowe, (2004), Snavely et al., (2008), Szeliski (2010) and Verhoeven, (2011).

1. A feature recognition algorithm is used to identify unique tie points across the image data set, e.g. scale invariant feature transform (SIFT) (Lowe, 2004).

2. A bundle block adjustment is performed to reconstruct camera positions and construct a sparse point cloud.

3. Ground control targets are identified within the scene and positioned on each individual photo in which they are visible. Placement accuracy of individual markers is within ~3cm (width of the X on targets).

4. The following step is iterated three times to reduce model error as estimated by the software. Sparse point cloud is optimised to fit ground targets. Points with estimated reprojection error >1.5cm are deleted. Repeat.

5. A dense point cloud is generated from the sparse point cloud model.

6. A 3D model, raster DSM and orthomosaics are output for analysis.

For this study DEM's were generated at 10cm pixel resolution and RGB orthomosaics were created at 5cm pixel resolution.

## 4.5 Accuracy Assessment

Accuracy of the DEM's is estimated in two ways. Firstly, the photogrammetric software provides a number of different error statistics including tie point matching error and GCP placement error (difference between actual coordinates and estimated position from SfM processing) (Table 4). These parameters describe how well the point cloud and camera positions fit the in scene ground targets. Error is provided as RMSE, xyz differences and pixel matching errors. These errors are reduced within the SfM workflow by optimising the point cloud while minimising potential position variance from control points. Outliers are selected and removed through a few iterations to reduce error estimates. Secondly, the surveyed GPS elevation at the check point locations is compared against the elevation extracted from the DEM surface providing a more precise estimate of absolute elevation error for each surface. A third method for estimation of error is to compare difference in the DSM elevations over regions that have experienced no change. Unfortunately no regions within the study area were identified that could be assured not to have undergone elevation change over the study period. This method was therefore not used.

## 4.6 Elevation Change, Surface Velocity and Orthomosaic Comparison

Analysis of glacier changes was conducted in ESRI Arc Map 10.2. Volumetric change was calculated for the entire survey area and the debris covered glacier section separately. This was done by clipping the DEM's to the glacier boundary and subtracting the 2014 DEM from the 2015 DEM; i.e. negative values equal elevation drop/ice loss. Surface velocity was





determined by manual feature tracking of 72 points on the glacier tongue to produce velocity vectors. Following vector creation a velocity surface was interpolated for the glacier tongue using the spline interpolation tool in Arc Map. To investigate melt patterns across the glacier surface horizontal movement must be removed so that the same area of the glacier surface is compared. To do this the velocity vectors above were used to orthorectify the 2015 orthomosaic to the same location as the 2014 orthomosaic. Vectors of ice cliff movement were then calculated between the two dates. Within the lake section of the survey area elevation changes of water bodies were measured and ice cored regions were investigated, as well as limited feature tracking to estimate horizontal movement.

## 5 Results and Discussion

### 5.1 GNSS and DEM Accuracies

Base station positions were located with an estimated combined (horizontal and vertical) error of 0.5cm (2014) and 7cm (2015) (Table 2). Positional errors of the GCP's and check points are all estimated to be under 2.5cm (2014) and 0.6cm (2015) (Table 2). Providing an estimated combined positional error of 3cm (2014) and 8cm (2015). Thus maximum expected errors in positions compared between the two dates are ~11cm. 2014 rover positions are less accurate due to the long (15km) baseline, however the longer base station occupation interval means they are overall better constrained. Rover positions in 2015 are more accurate due to a shorter (1km) baseline but the base station position is less accurate due to a much shorter occupation interval (6.5hrs versus almost 7 days) (Table 2). This could not be avoided as the base station could not be left in place overnight due to security concerns.

Estimated tie point (image) matching errors are summarised in Table 4, with a mean value of 0.63 pixels (at 3.5cm pixel size (Table 3)) for 2014 and 0.43 pixels (at 3.6cm pixel size (Table 3)) for 2015. Errors in both cases are sub pixel at the minimum pixel size; these results are improved when the DEM and orthomosaics are aggregated for analysis to 10cm and 5cm respectively. Additional error estimates are provided as the measured difference (in metres and pixels) between GCP coordinates and estimated GCP position within the SfM model (Table 4). These errors are sub centimetre for both years; 0.002m in 2014 and 0.003m in 2015. The error statistics provided above however tend to overestimate DEM accuracy. DEM accuracy should therefore be assessed by either comparing differences over no change areas or comparing to additional survey points not used in generating the DEM.

To gain a better understanding of the absolute accuracy of the two DEM surfaces elevations from the DEM were compared against those from check points surveyed in their respective years (Fig. 3, Fig. 4). The results of this analysis are summarised in Fig. 5 and Fig. 6. Mean difference between DEM and GNSS check points in 2014 is -0.037m with a standard deviation of 0.191m; in 2015 mean difference is 0.001m with a standard deviation of 0.044m. Negative values indicate underestimation of DEM surface with respect to surveyed positions and thus imply flattening of the DEM surface. Most errors are less than +/-



5cm, and are the combined result of errors in image matching/SfM processing, GNSS positional errors and DEM smoothing of small surface features. The 2014 data have a much larger error range than in 2015, however this is deemed to be the result of experimental error as opposed to true inaccuracies in the DEM surface. In 2014 check points were collected in areas of non uniform topography, i.e. atop rocks, between cracks etc, these minor topographic features are smoothed out of the DEM resulting in a larger difference between GNSS and DEM elevations. In 2015 check points were located in areas with only minor topographical variation within a roughly 0.5m radius therefore the impacts described above are minimised and estimated accuracy is higher.

**5.2 Glacier Tongue Surface Changes**

Changes in surface elevation were extremely variable across the study area, including areas of both loss and gain (Fig. 7). Over the glacier tongue (blue boundary Fig. 7) the mean elevation change was -0.75m from 23 July 2014 to 28 July 2015. This equates to 156,000 $m^3$ of ice loss. Standard deviation is 2.58m, with a minimum value of -18m and a maximum value of +11.5m, however from Figure 8 it is evident that almost all values are within -10m to +7.5m. Mean ice loss generally decreases moving up in elevation across the glacier tongue, and is greatest for elevations below 4530m, or roughly the lowest half of the glacier tongue. The largest elevation change of 18m of ice loss was measured at the calving face (Fig. 9, Fig. 10). From speaking with local climbing guides much of this loss occurred over a roughly two week period in March-April 2015. In the region behind the calving face considerable downwasting (roughly 1m to 8m) occurred likely as a result of collapse of the calving face, which had previously provided a buttressing effect for the ice behind.

Elsewhere on the glacier surface the most dramatic areas of surface ablation are associated with exposed ice cliffs and melt ponds (Fig. 10, Fig. 11). Exposed ice cliffs lack the insulation of thick debris cover and thus are exposed directly to intense incoming solar radiation and therefore experience rapid melting and horizontal recession across the glacier surface (Fig. 11) (Buri et al., 2016). Figure 11 shows the horizontal movement and expansion of these ice cliffs and the retreat of the calving face (note: the 2015 image has been georectified to remove any horizontal glacier movement based on calculated velocity vectors, see Fig. 7). Rates of horizontal retreat measured for these features range from 2m to 25m/yr with larger cliffs retreating more rapidly.

Appearance, expansion and disappearance of melt ponds is also visible in the imagery. Surface melt ponds have a lower albedo than exposed ice and therefore warm up rapidly; enhancing melt processes and increasing melt pond size. The disappearance/reduction of these features indicates potential subglacial drainage pathways through englacial conduits (Benn et al., 2012). Immerzeel et al., (2014) identified similar features through analysis of UAV datasets collected over a debris covered glacier in the Himalaya, and associated their presence with ice cliffs. In regions where lower amounts of ablation were measured this variability is likely associated with different debris thickness which provides either enhanced melt through



reduced albedo or reduced melt through insulation when debris thickness crosses a specific threshold (Brock et al., 2010; Reid et al., 2012; Reid and Brock, 2010).

The presence of ice cliffs and small surface depressions capable of forming melt ponds thus appear to be primary controls on melt processes of the glacier tongue. The fine scale of these surface features could not be resolved using medium and coarse resolution satellite imagery and would be extremely labour intensive to observe with in situ measurements. Debris thickness is likely a secondary control on melt rates for this glacier as the thick layer (>1m) of debris here primarily provides insulation as opposed to enhancing melt through albedo reduction (Brock et al., 2010; Reid and Brock, 2010).

Many sections of the glacier tongue show a gain in surface elevation (Fig. 7). These however are not zones of accumulation and can instead be explained by the highly irregular nature of the glacier surface. The glacier surface has numerous surface irregularities including topographic undulations, cliffs and large boulders. As the glacier moves downslope these surface features move through the scene thus elevation comparisons at an individual point are not actually comparing the same point on the glacier surface (which has moved) but instead compare the same position in real world coordinates. Therefore if a large boulder or surface feature moves down slope it will be measured as an increase in surface elevation at its new position. The inverse effect (elevation loss) can occur as large depressions move through the scene. Examples of this can be seen in Figure 10 where a large boulder roughly 10m high moves down slope producing a measured elevation decrease at is location in 2014 and a measured elevation increase at its location in 2015. Care must therefore be taken when comparing measurements of vertical change at a specific location. This is not an issue faced by non geodetic methods such as ablation stakes which move with the glacier and thus provide a measure of actual ablation at that position on the glacier surface. Vertical emergence may also contribute to the observed increases in ice elevation, however the glacier bed in the study section is likely fairly smooth and the glacier in this section undergoes no notable direction changes that could produce large emergence velocities. Therefore the impacts of emergence in creating these observed elevation increases is likely minimal.

### 5.3 Glacier Velocity

Surface velocities range from 27m/yr to 4m/yr from the upper glacier to the calving face respectively. The observed decrease in velocity down glacier is essentially linear. As the surface gradient is fairly constant over the glacier tongue (Fig. 7) this velocity reduction is likely due to a reduction in ice mass closer to the glacier terminus which is in part a result of the higher melt rates occurring over the lower sections of the glacier tongue. Very few measurements of glacier surface velocities are reported for the Cordillera Blanca. Emmer et al., (2015) reported maximum values of 11.2m/yr for a similar debris covered section at the much steeper Jatunraju glacier, which is less than half what is presented here. However the relative size of the accumulation area at Llaca is much larger than for Jatunraju. It is also possible that the glacier/bedrock interface is somewhat lubricated through contact with the proglacial lake, potentially increasing horizontal velocity.



## 5.4 Proglacial Lake Changes

Large changes were also measured within the proglacial lake system. The lake system can be divided into two parts. The lower section comprises the main (larger) lake and is the section for which the glaciology office has completed bathymetric surveys and volume assessments (Portocarrero, 2014b). This section was not surveyed and likely experiences limited change as its water level is controlled by the dam culvert. The upper section between the lake and the calving face includes numerous smaller ponds and sediments. Within this upper lake section numerous changes were observed over the survey period. Notably the appearance and disappearance of melt ponds, and areas of land subsidence (Fig. 12). Two locations experienced large elevation changes with surface lowering of 5-10m recorded over the study period (Fig. 12). These changes suggest that sediments within the lake system are interspersed with sections of remaining glacier ice which is slowly melting. As these ice remnants melt surface and subsurface hydrologic connections can open and close impacting the size and distribution of surface ponds and changing the surface topography.

Water level elevations for individual ponds within this section are not the same and are offset by up to 4m above the main lake which sits at 4480m, confirming that they are not connected directly to each other or the main lake. Drainage of these melt ponds is likely rapid as surface and subsurface channels open due to the melt of buried ice bodies and through the impacts of waves triggered by talus falls from the lateral moraine and calving of the glacier tongue. Low surface velocities were measured within this section ranging from 2.5m/year below the calving face to 0.5m/year at the end of the survey area where the main lake starts. These low velocities confirm that this section of the lake system remains connected to the main glacier.

Measured changes within the proglacial lake sediments illustrate that this system is actively evolving as a result of continued ice melt. As this process continues the extent and depth of the main lake will increase. Rapid drainage of perched melt water ponds some of which are considerably large (largest is 15,000m$^2$ and sits ~2.5m above main lake) within this upper lake section could produce rapid fluctuations in water inputs to the main lake, potentially impacting downstream infrastructure such as the moraine dam, power generator and water diversions. As the main lake expands northwards it will be in closer proximity to the steep headwalls of the valley which are potential locations for avalanche and rock fall debris that are triggers for GLOF hazards. UAV surveys provide a rapid, on demand method for continued monitoring of the evolution of this system.

## 6 Conclusion

This study deploys a custom designed hexacopter UAV to map dynamics of a debris covered glacier tongue in the Cordillera Blanca, Peru. To our knowledge it is the first time a UAV has been deployed for this purpose in the tropical Andes and is the highest altitude deployment of a multirotor UAV for mapping purposes in the current literature. We completed repeat aerial surveys (23 July 2014 and 28 July 2015) of the glacier tongue and proglacial lake system. Using a structure from motion workflow we generated highly accurate 10cm DEM's and 5cm orthomosaics. Analysis of these data reveals heterogeneous





changes in the glacier surface with some areas of the glacier losing as much as 18m of vertical elevation due to collapse of the calving face. Average glacier down wasting was -0.75m over the study period or roughly 156,000m$^3$ of ice loss. The most rapid areas of ice loss were associated with exposed ice cliffs and melt water ponds on the glacier surface. Surface velocities were recovered from manual feature tracking and ranged from 27m/yr in the upper surveyed section to around 2m/yr at the

calving face. Surface lowering and low surface velocities were also measured over the sediments within the proglacial lake indicating that extensive regions of glacier ice remain buried within the lake sediments that are still connected horizontally to the glacier tongue. Only limited horizontal retreat of the glacier tongue was recorded thus simple measurements of changes in aerial extent can provide only a limited understanding of actual changes in glacier ice. Continued down wasting of the glacier surface and melting of buried ice within the proglacial lake sediments will increase the size of the proglacial lake over time.

As the lake boundary migrates towards the valley headwall GLOF risks associated with avalanches from the cirque above are likely to increase.

This study achieves two primary goals. Firstly, it clearly illustrates the viability and suitability of this technology for studying high resolution changes in mountain glaciers, and the feasibility of operating multirotors at high elevation (>5000m). Secondly

it provides insights in to the processes that control glacier melt over a debris covered tropical glacier. Our findings regarding the importance of ice cliffs and surface melt ponds in debris covered glacier melt processes corroborate findings for debris covered glaciers in the Cordillera Blanca (Emmer et al., 2015) and the Himalaya (Buri et al., 2016; Immerzeel et al., 2014). The high spatial resolution of UAV derived datasets facilitates a much better understanding of surface heterogeneity and spatial variability in debris covered glacier melt processes and changes in proglacial lake systems. UAV's offer a low cost on demand

alternative to existing technologies, they can provide unique insights into dynamic earth system processes and are ideal for small scale studies in complex mountain regions.

**Acknowledgements**

This study was co-funded by NSF Grants: BCS-1010550 and BCS-1434248, The American Philosophical Society, The American Geographical Society, The Explorers Club, The Geological Society of America and The Ohio State University

Office of International Affairs. Additional support in the form of GNSS equipment loan for 2014 was provided by the UNAVCO Facility with support from the National Science Foundation (NSF) and National Aeronautics and Space Administration (NASA) under NSF Cooperative Agreement No. EAR-0735156. Permission to operate the UAV within the Huascaran National Park was provided by the Parque Nacional Huascaran Office in Huaraz.





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



**Tables**

| Survey Year | Ground Target Used | Number of Targets | Number of Check Points | Base Station | Rover |
|---|---|---|---|---|---|
| 2014 | 10 inch diameter plastic plates; sprayed florescent yellow, silver taped X for centre | 14 | 18 | Trimble 5700 L1/L2 Receiver; Trimble Zephyr Geodetic Antenna (UNAVCO) | Topcon GRS1 L1 Receiver, Topcon PG-A1 External Antenna |
| 2015 | 12 inch square floor tiles; side a: yellow coroplast pink X centre; side b: black and white quads | 11 | 13 | Topcon Hiper SR L1/L2 Integrated Receiver and Antenna | Topcon GRS1 L1 Receiver, Topcon PG-A1 External Antenna |

**Table 1: Llaca glacier GNSS ground control survey and hardware specifications.**

| GNSS Survey Error Estimates | July 23, 2014 | July 28, 2015 |
|---|---|---|
| Base Station Observation Length | >106 Hours | 6.5 Hours |
| Rover Observation Length | 5 mins | 5 mins |
| Base Station and Rover Observation Rate | 1 Hz | 1 Hz |
| Average Horizontal Baseline Length | 14.9 km | 1.2 km |
| Average Vertical Baseline Length | 1400 m | 9 m |
| Base Station Horizontal Confidence (95%) | 0.0022 m | 0.0326 m |
| Base Station Vertical Confidence (95%) | 0.004 m | 0.041 m |
| Total GCP Horiz. Er. Estimate (Rover RMS + Base Station) | 0.017 m | 0.035 m |
| Total GCP Vert. Er. Estimate (Rover RMS + Base Station) | 0.019 m | 0.045 m |
| Total Check Pt Horiz. Er. Estimate (Rover RMS + Base Station) | 0.011 m | 0.035 m |
| Total Check Pt Vert. Er. Estimate (Rover RMS + Base Station) | 0.020 m | 0.044 m |

**Table 2: Estimated GNSS survey positional errors for use in SfM ground control and absolute DEM error analysis. Note: Total error estimates include base station and rover error estimates.**

| | July 23, 2014 | July 28, 2015 |
|---|---|---|
| **Number of Images Captured** | 601 | 929 |
| **Number of Images Used** | 323 | 826 |
| **Av. Flying Altitude** | 110.46 m | 116.11 m |
| **Ground Resolution** | 0.0353 m/pixel | 0.0367 m/pixel |
| **Coverage Area** | 0.489 km$^2$ | 0.542 km$^2$ |


**Table 3: UAV Photogrammetric survey parameters.**



| | July 23, 2014 | July 28, 2015 |
|---|---|---|
| **Tie Points** | 84530 | 53545 |
| **Projections** | 387568 | 357433 |
| **Tie Point Matching Error** | 0.632 pixels | 0.430 pixels |
| **GCP X Error** | 0.0009 m | 0.0019 m |
| **GCP Y Error** | 0.0007 m | 0.0017 m |
| **GCP Z Error** | 0.0012 m | 0.0005 m |
| **GCP Total Error** | 0.0016 m | 0.0026 m |
| **GCP Total Error** | 0.0809 pixels | 0.0231 pixels |
| **Max DEM Resolution** | 0.071 m/pixel | 0.073 m/pixel |
| **Point Cloud Density** | 200 points/m$^2$ | 185 points/m$^2$ |

**Table 4: Estimated pixel matching and DEM reconstruction errors from SfM processing workflow.**

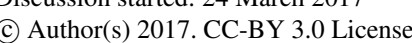

**Figures**

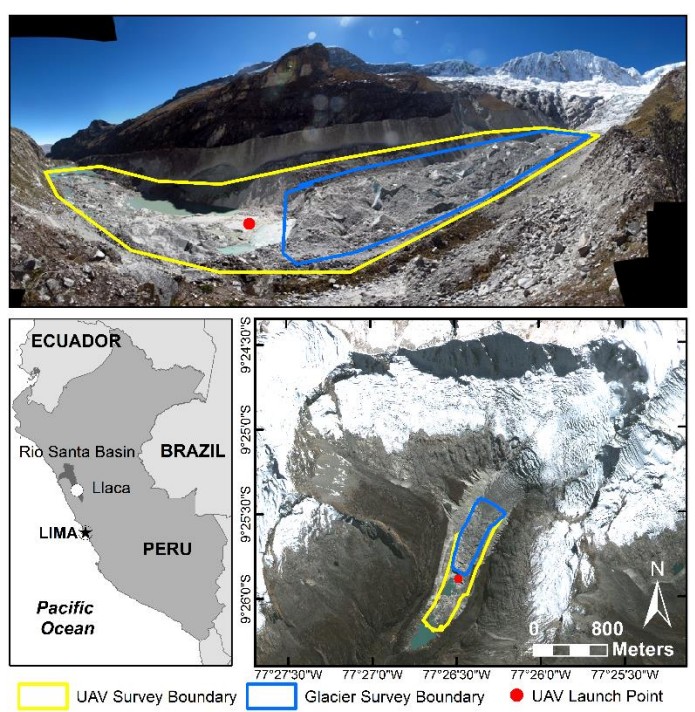

**Figure 1: Llaca glacier location map within the Cordillera Blanca. Showing UAV launch point and survey area boundaries. UAV survey boundary (yellow line) is maximum extent of study area covered by both 2014 and 2015 surveys; Glacier survey boundary (blue line) is maximum extent of glacier tongue covered by both 2014 and 2015 surveys.**

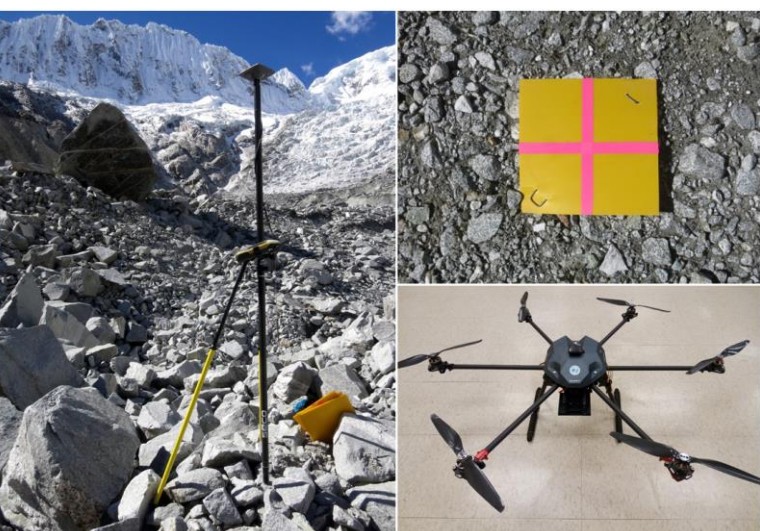

**Figure 2: GNSS survey at Llaca glacier (left), installed ground control target (2015) (top right), hexacopter UAV platform used in this study (bottom right).**





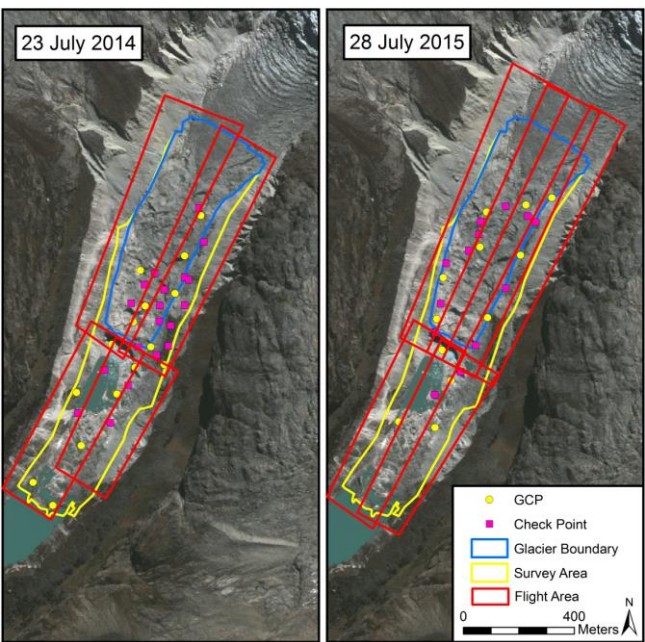

**Figure 3: Flight coverage areas, GCP's and check point locations for 2014 and 2015 UAV survey campaigns. Background imagery from Worldview 13 July, 2011.**

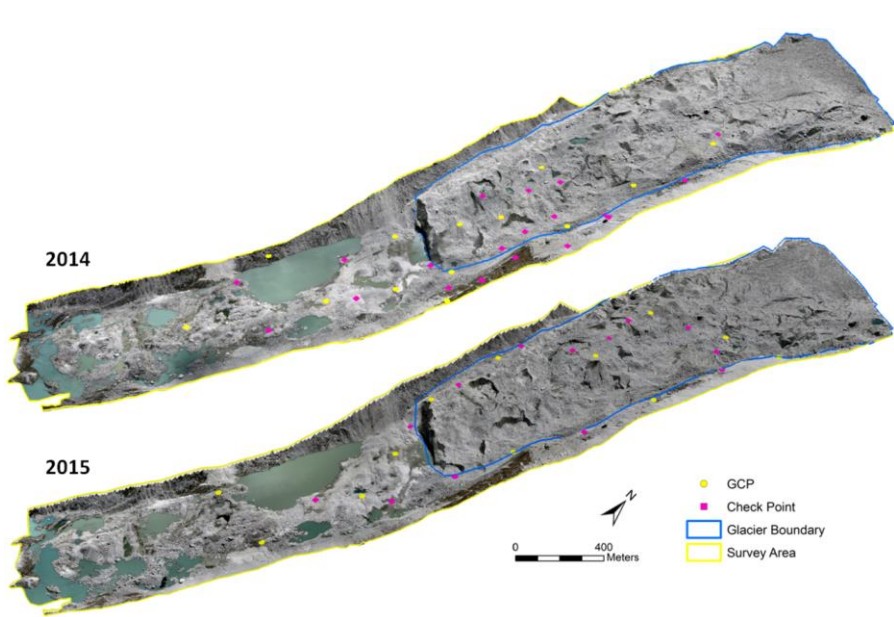

**Figure 4: Photogrammetric DEM's and RGB orthomosaics of Llaca glacier tongue in 2014 (top) and 2015 (bottom), constructed through SfM workflow.**




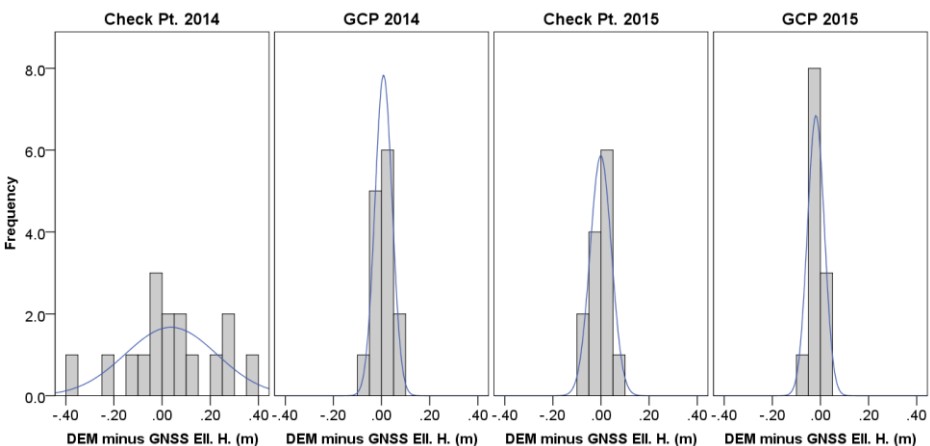

**Figure 5: Histogram plot of elevation differences (errors) between check points and GCP surveyed elevations and DEM elevations for 2014 and 2015 respectively Note: negative values indicate underestimation of DEM surface with respect to surveyed positions and thus imply flattening of the DEM surface. Most errors are less than +/-5cm and all within +/-10cm except for 2014 check points. Large 2014 check point errors are assumed to be the result of poor location selection.**

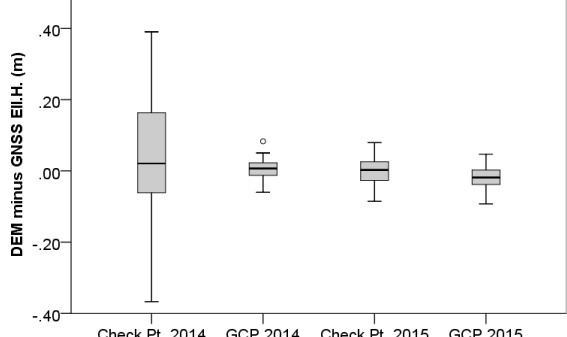

**Figure 6: Box and whisker plot of elevation differences (errors) between check points and GCP surveyed elevations and DEM elevations for 2014 and 2015 respectively. Thick lines represent median value, boxes show upper and lower quartiles and whiskers indicate minimum and maximum values, circles are outliers. Note: negative values indicate underestimation of DEM surface with respect to surveyed positions and thus imply flattening of the DEM surface. Most errors are less than +/-5cm and all within +/-10cm except for 2014 check points. Large 2014 check point errors are assumed to be the result of poor location selection.**



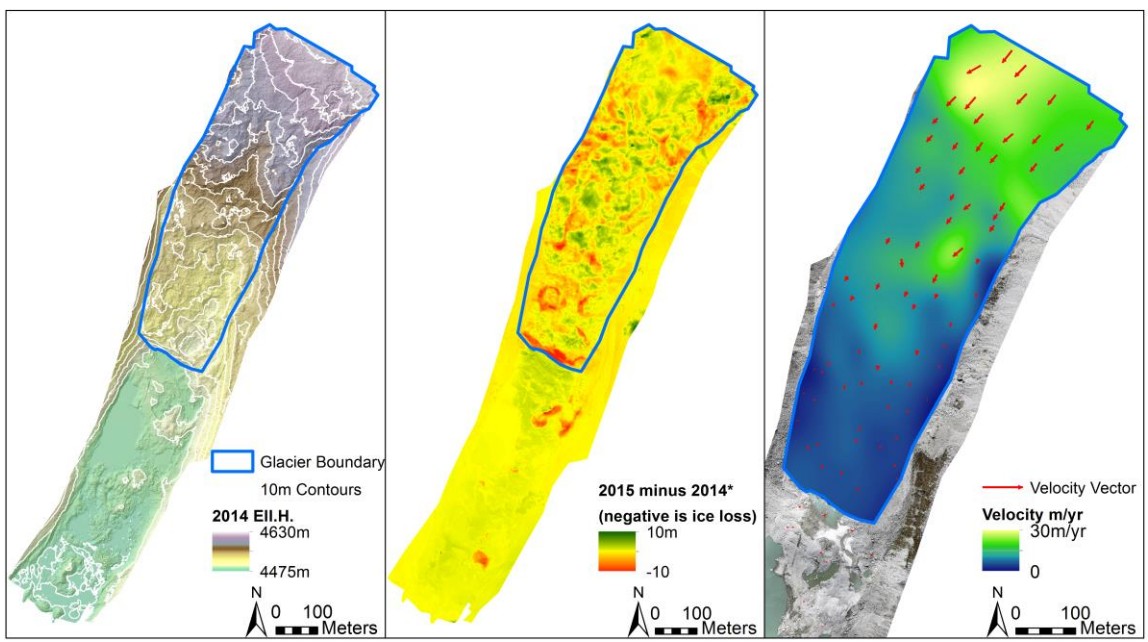

**Figure 7: 2014 SfM DEM elevations (Ell.H.) (left); Vertical change in elevation from 23 July 2014 to 28 July 2015 (centre); Glacier surface velocity tracks and interpolated velocity surface (right). *Note: centre panel values are truncated to +/-10m for display purposes (see Figure 8).**

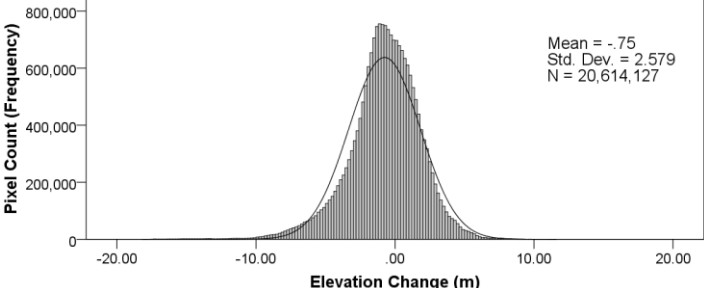

**Figure 8: Histogram of measured elevation changes over glacier tongue (blue boundary in Figure 7) from 23 July 2014 to 28 July 2015. Calculated from subtraction of 2014 DEM from 2015 DEM (i.e. negative values are ice loss) at 10cm DEM pixel size.**

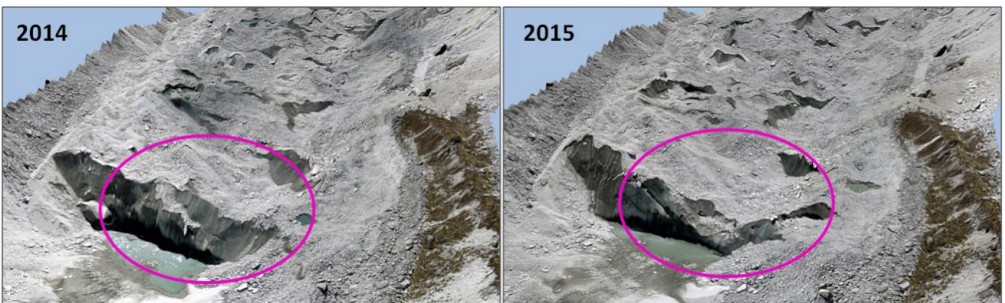

**Figure 9: Collapse of the calving face.**




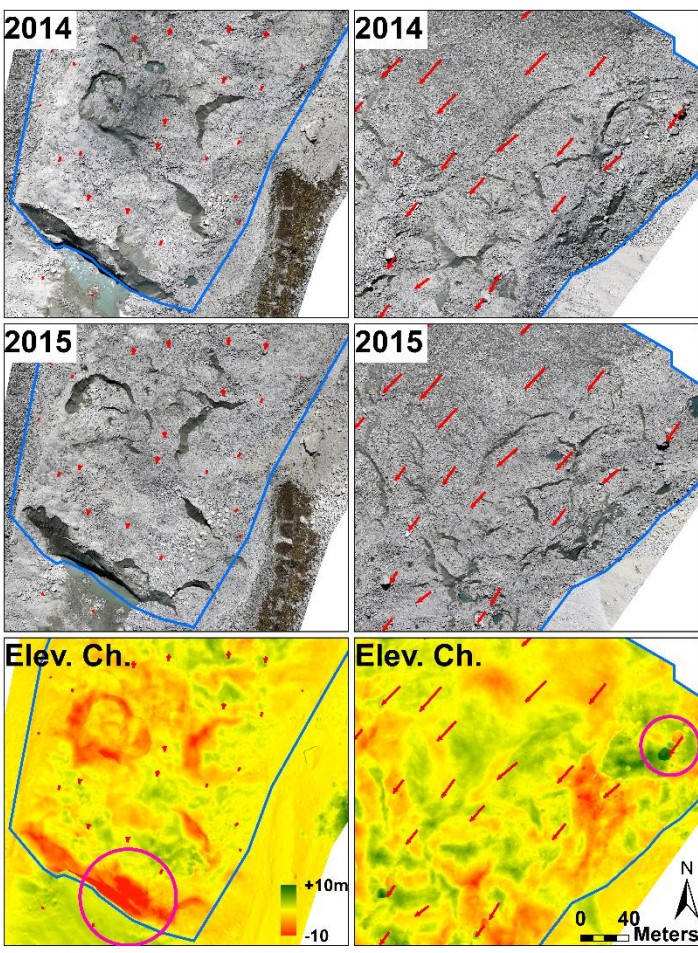

**Figure 10: Locations of maximum and minimum change in ice elevation on Llaca glacier tongue.Upper row panels show 2014 orthomosaic; middle row panels show 2015 orthomosaic; lower row panels show elevation change where 2014 elevation is subtracted from 2015 elevation (i.e. negative value is ice loss). Maximum ice loss of 18m occurred at the calving face (pink circle lower left panel). Maximum gain of 11m is recorded in upper right section where a large boulder (roughly 10m tall) has moved through the scene (visible by comparison of 2014 and 2015 orthomosaics). Blue line is glacier tongue boundary, red arrows are velocity measurements. Note: elevation change values are truncated to +/-10m for display purposes.**





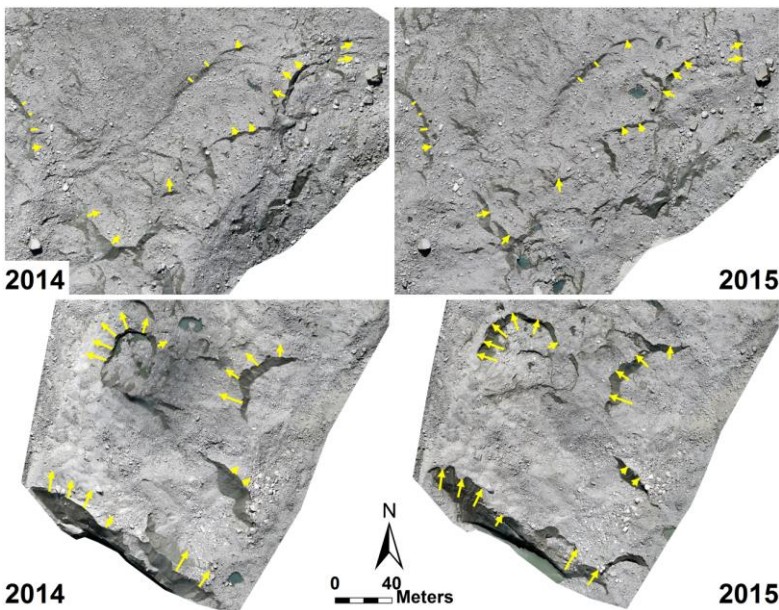

**Figure 11: Rapid ice loss around exposed ice cliffs and surface melt ponds.Top row images are from upper right of surveyed area, lower row images are behind calving face. Yellow arrows indicate horizontal movement of feature from 2014 to 2015 of ice cliff position; 2-25m/year. More rapid retreat of ice cliffs is observed in the lower glacier, where cliffs are larger and temperatures are warmer. NOTE: 2015 images have been shifted (georectified) to 2014 positions based on velocity vectors (Figure 7), i.e. horizontal glacier movement is removed so that the same location relative to glacier surface is compared.**

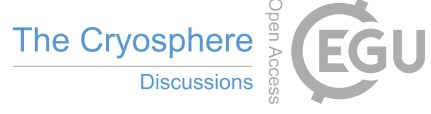

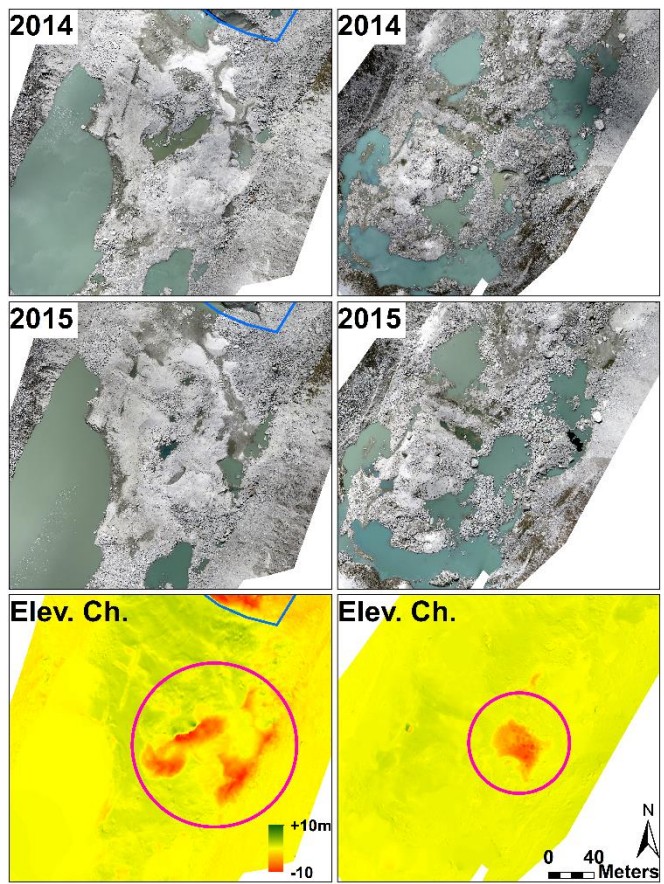

**Figure 12: Locations of major elevation change (ice loss) within lake sediments (pink circles), indicate buried glacier ice is still present. Low surface velocities were also measured here (~2.5m/yr below calving face in left panels, and ~0.5m/yr at end of surveyed region in right panels, indicating these sediments are still connected to the main glacier tongue. Upper row panels show 2014 orthomosaic; middle row panels show 2015 orthomosaic; lower row panels show elevation change where 2014 elevation is subtracted from 2015 elevation (i.e. negative value is ice loss). Blue line is glacier tongue boundary. Note: elevation change values are truncated to +/-10m for display purposes.**