# Peer review of "Monitoring Tropical Debris Covered Glacier Dynamics from High Resolution Unmanned Aerial Vehicle Photogrammetry, Cordillera Blanca, Peru"

_The Cryosphere, 2017_

## Short Comment (SC1) · 2 Apr 2017

Wigmore and Mark (2017) provide a detailed review of the methodology employed using a UAV for repeated mapping of the debris tongue of Llaca Glacier, Peru. The emphasis is on the methodology and I found this to be comprehensive and comprehendible. The level of detail provided by the UAV photogrammetry generated detailed maps of velocity and surface elevation change indicating the value of the approach. There is less emphasis on detailed review of the results. More attention should be given to the speed of supraglacial lake expansion and the volume of water they contain

in Section 5.4, as this is both important but also another measure of the utility of the UAV. In Section 5.2 additional discussion of any ablation rates that could be derived in areas of low velocity such as the right panels in Figure 10 such as a zone mean would be useful. Below are specific comments, which are generally minor.

2-14: Differentiate Cordillera Blanca from Himalaya ie. The Himalayan glaciers are in a warmer/wetter environment.

3-10: Is this paragraph needed? Full glacier mass balance is discussed, but this study is not completing full glacier mass balance.

4-5: To improve the utility of one factor.. ...

4-30: Remove this sentence since you have sufficient references without and there are so many videographers uses of UAV's . . . "At higher elevations M. Willis (pers. com.) and S. Wernke (pers. com.) have had success using multirotor and fixed wing platforms for archaeological mapping in the Ecuadorian and Peruvian Andes respectively, at altitudes of up to 4000m. . .

5-26: Reference a specific glacier such as Vallunaraju which shares a divide and terminates at ∼4750 m.

6-6: The lake cannot be classified as directly above Huarez which is over 12 km from the glacier. The drainage channel also enters the Rio Santa downstream of the main city of Huarez in the northern reaches of the city.

10-11: How does the ice loss compare to net annual ablation in this area? If unknown report that.

10-20: Any temperature records from near this portion of the glacier during melt season?

10-28: Any observation on melt pond albedo or water temperature. The overall water temperature statement is not applicable if the surface pond is connected to surface

streams that lead to rapid turnover.

10:28: You note the change in position of cliffs and melt rates of such features. What about the rate of melt pond expansion? This is commented upon and would be important to note.

11-27: What about the impact of surface slope on the velocity gradient, there is close to twice the slope in the upper study area, as in the lower area of the study?

11-32: There is sufficient melt for lubrication of any of the lower reaches of these glaciers during the melt season. In this case there is likely partial flotation of the lower section of the glacier, which is less lubrication than having a drainage system that is full and hence higher basal water pressure.

12-23: Likely lake volume range?

---

## Referee Comment (RC1) · C. Huggel (Referee) · 5 May 2017

General comments:

This is a paper documenting a pioneering UAV based study in the Cordillera Blanca in Peru. While UAV studies have become widespread over the past few years still very few applications exist on high elevation glaciers, and only a couple of them to date in the tropical Andes. This is among other related to the challenges of operating a UAV at elevations higher than 4000 m or even 5000+ m. The technical (UAV related) part of this paper is basically ok in my opinion. The question is how much insight the UAV data

provides in terms of understanding of the glaciological processes. In general I think the paper is stronger on the technical side than on the process side. This is also due to the limited temporal and spatial coverage of the UAV data, i.e. one repeat flight, 1 year after the other, and focusing on the terminus area of the glacier. This limitations is totally understandable considering the challenges of operating a UAV in these conditions, on top of further requirements for high-precision geodetic measurements. And I would re-iterate the merit of this study to provide high-resolution information on a debris covered glacier terminus in the tropical Andes where such studies are very rare or missing. However, both limited spatial and temporal coverage implies some constraints: 1 year is little to analyze characteristic patterns on a glacier, and the lack of information for the upper glacier areas limits the process understanding as it needs to ignore the dynamics of the entire glacier. The paper should more explicitly acknowledge these limitations.

Concerning the research questions (RQ) I think they are ok but RQ 3 and 4 are not fully addressed. The process of glacier change should be identified more clearly. E.g. can the effect of ice cliff related mass loss be better quantified? Some more evidence may be provided concerning the proposal that the ice cliffs are the primary control on mass loss. A comparison to other mass loss information on glaciers in the Cordillera Blanca could be useful. I'm furthermore unclear whether the large share of positive elevation change is just related to blocks and cliffs and other objects moving through the scenes?

RQ4 is not addressed in much detail in the paper. E.g. much of the connectivity issue remains vague. I suggest to clarify which connectivity is exactly in place here: hydrological connectivity? Can this be addressed using UAV data? The text on future lake changes is not informed by UAV data. Does it tell us more than we know from other debris covered glaciers? E.g. the time frame?

I suggest some major revisions and re-organization of sections 1 and 2. The current organization is rather odd, or at least unusual, not supporting clarity. The research gaps and objectives should be better worked out. An option is that section 1.1 and 1.2

go in a separate background section after the Introduction. Section 2 could then be integrated in the Introduction.

Overall, I think the paper is mostly fine on the technical side but needs to be enhanced on the glacier process side, i.e. interpretation of the UAV data to the extent possible, comparison with available data from other glaciers in the region or possibly outside, and further dig into literature supported relevant processes. It may be interesting to be more explicit on the potential and limitations of UAV methods for fostering understanding of glacier change. Certainly, UAV's role for glacier monitoring is strong and increasing. But the authors claim that this data is providing unprecedented insights into glacier processes and this needs to be further elaborated and demonstrated. I'm confident that the authors can deliver a valuable and impact-full paper after some major revisions.

Detailed comments:

Page 2, line 1: rise in average glacier terminus elevation? 2,7: Huggel et al should be 2003 2, 19 ff.: I don't think too many people would agree on the distinction of four primary methods to quantify glacier change. A more classical distinction foresees a distinction of glaciological, geodetic and hydrological methods to determine glacier mass change. I think the four methods identified here are somewhat odd, or arbitrarily identified (e.g. I would not see energy balance modeling as a method to determine glacier mass change). This also relates to page 2, 29 which is an incorrect statement in my opinion. 3,4: you may add that energy balance studies are typically limited to point measurements of a weather station. 3, 24: 10-100 m rather than decimeter (and vertical accuracy may also be indicated with a range) 5, 2: there are other recent studies using UAV in the tropical Andes but they are not yet published, at least to my knowledge. 7, 3: spell out SfM the first time used 7, 22: do you mean outdated for use…? 10, 10: I suggest to compare measured elevation changes to other available measurements in the Cord. Blanca. 11, 25 ff: Section 5.3 is very thin. How compare these velocities to other debris covered glaciers? What are the implications? 13, 14: did the UAV fly over 5000m? In the method section it is mentioned that flight height

was 100 m above ground which would translate into a max of ca. 4800 m asl. In case you have experiences flying over 5000 m it may be interesting to know and report (but be specific). 13, 15: the insights into the processes that control glacier melt could be enhanced in the main body of the text (see above comment).

List of references: please check. Some references are duplicated.

Figure 7: how was the glacier boundary determined? This is often difficult on debris covered glaciers. It would be helpful if the location of figures 10, 11, 12 could be indicated in Fig. 7. The color scale should be improved: especially the 0 value level could be made more clearly visible.

Christian Huggel, University of Zurich

---

## Referee Comment (RC2) · Anonymous Referee #2 · 30 May 2017

The paper presents UAV survey results from the Cordillera Blanca, Peru. The data are novel, in that they provide information on a comparatively little-studied area and add to the limited number of UAV surveys on high-altitude glaciers. However, for me, the study does not really address its objectives. It measures changes over a portion of the glacier, but does not capture the full volume change. I also question why the velocities were determined manually and for 72 points, rather than using more spatially comprehensive feature tracking. For objective three, the study does not explicitly investigate the role of debris thickness and its links to melt: it identifies debris as a secondary control, because it is thick at the terminus, but does not look at smaller scale varia-

tions or assess its impact on melt rates at higher elevations. It also does not consider that the debris may well be the reason for the tongues existence and characteristics (e.g. presence of ice cliffs). Overall, the discussion of debris cover is a little simplistic. Similarly, I felt Objective 4 was not properly addressed, and was a more limited discussion of the expansion of melt pools between two surveys, with some speculation about the potential for rapid drainage and lake expansion. I'm not sure how much this section extends our understanding of the interaction between the glacier and the lake. Overall, the paper has some valuable and interesting data, but it needs to be more specific about what the data can actually tell us (in terms of glacier dynamics, ice loss, lake interactions etc.) and the objectives should be re-focused accordingly. It would be good to see more comparison of these processes (e.g. melt cliffs, pools, lake formation) with analogues in the Himalaya: we have quite a bit of data and process knowledge from there, so it would be could to link these results to this literature and to make the comparison between different regions and sizes of glacier. My detailed comments are below. Page 1 Line 10: What about thinning, as well as retreat? Also, altering THE timing. Line 19: Why the only the tongue and not the whole glacier? This would be important for water resources. Line 23: I don't follow how this shows a continued connection to the glacier tongue: the pieces of detached ice could just be melting away in-situ, completely dynamically detached from the glacier. Line 27: First statement needs a reference and give % for the Cordillera Blanca. Page 2 Line 1: Rise in glacier terminus elevation sounds odd. Line 8: is>are Line 13: what is meant by 'geomorphic change for a debris covered glacier'? Do you mean glacier change, e.g. shrinkage / retreat? Line 19: I'm not sure this paragraph is needed. It's pretty general. I think it would be better to just say which approach was used and why, maybe at the start of the methods. I feel like all of section 1.1. is stepping too far back in terms of explanation and would be much better if it cut out the general material about each method, and focused specifically on its previous application to the study area (or relevant comparison glaciers, e.g. debris-covered glaciers) or why it hasn't been used previously. The paper does link to the study area at points, but it would benefit from

a tighter, more focused argument throughout. Page 5 Line 4: I like that the objectives are clearly stated. Line 15 & 16. Reference if possible. Line 26: How was this determined? By this study or previous work? Needs to be cited if it is the latter. Page 7 Line 22: dated for use? Line 28: Was this why only the lower portion of the glacier was surveyed? Page 8 Line 16: DEMs (remove the ') Page 9 Line 1: Why do this manually? Why not use e.g. Cosi-Corr or Imgraft? This should give you a much more spatially extensive velocity field, so you don't need to interpolate. The glacier is quite small, but 72 points does not seem a lot when you have a heterogenous surface (pools, cliffs, debris cover). Based on Line 5, I'm not clear if these offsets were then used to calculate the distance the cliffs were advected by the flow or to determine how much they had melted back (i.e. by taking out the velocity component and then subtracting the two DEMs). If the data are being used to estimate ice cliff melt rates, then can we really accurately separate out the local flow velocity from the melt rates (which are likely to be a few tens of cm, to a couple of meters) using just 72 points? Page 10 Line 12: For me, this shows a limitation to addressing objective 2 (and 4, to a lesser extent). We can't assess total volume changes or its potential future evolution, without accounting for the entire glacier. Line 23: As noted above, how confident can we be of removing the horizontal velocities local to the ice cliff, when 72 points were used across the glacier? Line 27: Needs a reference. Line 31: Quite vague and speculative. Do you have any data on debris thickness? Page 11 Line 6: I don't think you can say debris cover is secondary here. First, the differential melt associated with ice cliffs is at least partly due to the removal of the debris layer. Second, the presence of the thick debris layer may well be why this tongue still exists at these altitudes (you note earlier in the paper this is a comparatively low altitude for the region), so although it doesn't increase melt rates, it is still an important control on them. There are no in-situ measurements of melt versus debris thickness, so I think it is hard to make this statement, particularly as the study only focuses on the tongue where the debris cover is thick. Debris may well be important here for supressing melt, then accelerate melt further up. Line 10: I'm unclear how much of the change discussed earlier in 5.2. is due to this movement

of large objects, versus net change. We need to distinguish the two for e.g. forecasting water resources. Line 32: How? It would be tricky for the lake water to get very far up glacier. Could also relate to basal topography. Figure 4: I would make the dots bigger, so they are easier to see. A black outline would help. Figure 7: Make the scale bards categorised, rather than stretched, as it's easier to read of individual values. Figures 9-12. It would be useful to have some context about where these sites are, e.g. using extentboxes on Fig. 1 or 2.

---

## Editor Comment (EC1) · C. R. Stokes (Editor) · 12 Jun 2017

Dear Oliver,

You will, by now, have received formal notification that the open discussion of your manuscript has closed.

I would like to thank Christian Huggel, the anonymous referee and Mauri Pelto for their comments on your manuscript. All three sets of comments support publication of the manuscript, but they also identify a number of issues that would need to be addressed

in a revised manuscript, and which would likely constitute major revisions. A common theme is the rather unbalanced nature of the manuscript, with too much background, some objectives that are not fully-addressed, and an emphasis on the methods, rather than a more detailed discussion of the results and their implications. The latter would clearly increase the impact of your work and is something I would like to see addressed for publication in this journal.

Indeed, my own reading of the manuscript is that the study has clear potential to be published in The Cryosphere, but I would like to see some of the key structural issues addressed. I would recommend that sections 1, 1.2 and 2 are merged into a concise introduction (e.g. the summary of methods of identifying glacier change is unnecessary), and you might want to consider re-visiting the four objectives (e.g. perhaps set up a broader aim or fewer number of objectives that you specifically address). I would then recommend that Section 5 (Results and Discussion) is organised into a more orthodox structure that details the results and then presents some discussion in a separate section. This re-structuring should allow you to address many of the questions/issues raised by the reviewers and more clearly expand on some of the key results. Whilst I feel strongly that these structural issues must be addressed, and the manuscript would benefit from a clearer discussion of some of the key results, there may be a few instances where you do not necessarily want to broaden the manuscript and over-reach the results. This would be acceptable, but I would require a detailed rebuttal/justification.

If you have any queries, please do not hesitate to get in touch.

Kind regards,

Chris Stokes Editor

---

## Author Comment (AC1) · 7 Jul 2017

The authors would like to thank Mauri Pelto who took the time to provide a detailed short comment response to this manuscript. Pelto's constructive comments and suggestions will greatly improve the quality and impact of the manuscript. Our responses can be found below a reiteration of the comments.

Wigmore and Mark (2017) provide a detailed review of the methodology employed using a UAV for repeated mapping of the debris tongue of Llaca Glacier, Peru. The

[Figure]

emphasis is on the methodology and I found this to be comprehensive and comprehendible. The level of detail provided by the UAV photogrammetry generated detailed maps of velocity and surface elevation change indicating the value of the approach. There is less emphasis on detailed review of the results. More attention should be given to the speed of supraglacial lake expansion and the volume of water they contain in Section 5.4, as this is both important but also another measure of the utility of the UAV.

Response: We will expand the discussion of our results and specifically address the likely role of supraglacial lakes, as these appear to be a critical driver of melt on the glacier tongue. We will provide an estimate of melt rate expansion between the two survey dates, however estimating their volume is not possible with our current dataset.

In Section 5.2 additional discussion of any ablation rates that could be derived in areas of low velocity such as the right panels in Figure 10 such as a zone mean would be useful.

Response: We originally derived zonal statistics by elevation band (10m intervals) across the glacier tongue (see figure below). These were removed in the final edit of the manuscript in the interest of space. However, with a reduction of the introduction and expansion of the results and discussion these could be included.

Below are specific comments, which are generally minor. 2-14: Differentiate Cordillera Blanca from Himalaya ie. The Himalayan glaciers are in a warmer/wetter environment.

Response: Agreed.

3-10: Is this paragraph needed? Full glacier mass balance is discussed, but this study is not completing full glacier mass balance.

Response: We will likely delete this paragraph when the introductory sections are consolidated per general response.

4-5: To improve the utility of one factor: : :..

Response: Agreed

4-30: Remove this sentence since you have sufficient references without and there are so many videographers uses of UAV's : : : "At higher elevations M. Willis (pers. com.) and S. Wernke (pers. com.) have had success using multirotor and fixed wing platforms for archaeological mapping in the Ecuadorian and Peruvian Andes respectively, at altitudes of up to 4000m: : :

Response: Agreed.

5-26: Reference a specific glacier such as Vallunaraju which shares a divide and terminates at _4750 m. Response: Agreed.

6-6: The lake cannot be classified as directly above Huarez which is over 12 km from the glacier. The drainage channel also enters the Rio Santa downstream of the main city of Huarez in the northern reaches of the city.

Response: Agreed, we will correct/reword appropriately.

10-11: How does the ice loss compare to net annual ablation in this area? If unknown report that.

Response: It may be difficult to find a value that is comparable as the area surveyed was only the glacier tongue – and it is debris covered. However, we will see if there is something appropriate to use and include in the discussion if possible.

10-20: Any temperature records from near this portion of the glacier during melt season?

Response: Yes, we have a T/RH sensor installed above the glacier tongue (South moraine).

10-28: Any observation on melt pond albedo or water temperature. The overall water temperature statement is not applicable if the surface pond is connected to surface streams that lead to rapid turnover.

Response: Pond albedo is lower than exposed ice cliffs. Unfortunately no water temperature measurements were made. In general the supraglacial melt ponds appear to receive inflow from surface streams, and melt from surrounding ice cliffs but have no visible surface outflow, thus turnover is not rapid.

10:28: You note the change in position of cliffs and melt rates of such features. What about the rate of melt pond expansion? This is commented upon and would be important to note.

Response: We will provide an estimate of rate expansion for the supraglacial ponds.

11-27: What about the impact of surface slope on the velocity gradient, there is close to twice the slope in the upper study area, as in the lower area of the study?

Response: We will include this.

11-32: There is sufficient melt for lubrication of any of the lower reaches of these glaciers during the melt season. In this case there is likely partial flotation of the lower section of the glacier, which is less lubrication than having a drainage system that is full and hence higher basal water pressure. 12-23: Likely lake volume range?

Response: For the upper lakes/ponds included in this survey the volume is unknown. Bathymetric surveys were completed for the lower main lake in 2004, with a max depth of 17m and total volume of 274,000m3.
* * *
[Figure]

**Fig. 1.** Llaca elevation band stats

---

## Author Comment (AC2) · 7 Jul 2017

The authors would like to thank Christian Huggel who took the time to provide a detailed review of this manuscript. Huggel's constructive comments and suggestions will greatly improve the quality and impact of the manuscript. Our responses can be found below a reiteration of the comments.

General comments: This is a paper documenting a pioneering UAV based study in the Cordillera Blanca in Peru. While UAV studies have become widespread over the

[Figure]

past few years still very few applications exist on high elevation glaciers, and only a couple of them to date in the tropical Andes. This is among other related to the challenges of operating a UAV at elevations higher than 4000 m or even 5000+ m. The technical (UAV related) part of this paper is basically ok in my opinion. The question is how much insight the UAV data provides in terms of understanding of the glaciological processes. In general I think the paper is stronger on the technical side than on the process side. This is also due to the limited temporal and spatial coverage of the UAV data, i.e. one repeat flight, 1 year after the other, and focusing on the terminus area of the glacier. This limitations is totally understandable considering the challenges of operating a UAV in these conditions, on top of further requirements for high-precision geodetic measurements. And I would reiterate the merit of this study to provide high-resolution information on a debris covered glacier terminus in the tropical Andes where such studies are very rare or missing. However, both limited spatial and temporal coverage implies some constraints: 1 year is little to analyze characteristic patterns on a glacier, and the lack of information for the upper glacier areas limits the process understanding as it needs to ignore the dynamics of the entire glacier. The paper should more explicitly acknowledge these limitations.

Concerning the research questions (RQ) I think they are ok but RQ 3 and 4 are not fully addressed. The process of glacier change should be identified more clearly. E.g. can the effect of ice cliff related mass loss be better quantified? Some more evidence may be provided concerning the proposal that the ice cliffs are the primary control on mass loss. A comparison to other mass loss information on glaciers in the Cordillera Blanca could be useful. I'm furthermore unclear whether the large share of positive elevation change is just related to blocks and cliffs and other objects moving through the scenes? RQ4 is not addressed in much detail in the paper. E.g. much of the connectivity issue remains vague. I suggest to clarify which connectivity is exactly in place here: hydrological connectivity? Can this be addressed using UAV data? The text on future lake changes is not informed by UAV data. Does it tell us more than we know from other debris covered glaciers? E.g. the time frame?
[Figure]

I suggest some major revisions and re-organization of sections 1 and 2. The current organization is rather odd, or at least unusual, not supporting clarity. The research gaps and objectives should be better worked out. An option is that section 1.1 and 1.2 go in a separate background section after the Introduction. Section 2 could then be integrated in the Introduction.

Overall, I think the paper is mostly fine on the technical side but needs to be enhanced on the glacier process side, i.e. interpretation of the UAV data to the extent possible, comparison with available data from other glaciers in the region or possibly outside, and further dig into literature supported relevant processes. It may be interesting to be more explicit on the potential and limitations of UAV methods for fostering understanding of glacier change. Certainly, UAV's role for glacier monitoring is strong and increasing. But the authors claim that this data is providing unprecedented insights into glacier processes and this needs to be further elaborated and demonstrated. I'm confident that the authors can deliver a valuable and impact-full paper after some major revisions.

Response: We thank you for your feedback and agree that the limitations of these data should be more explicitly addressed with respect to what a limited spatial extent (only the tongue) and temporal range (two dates) can tell us about glacier processes.

Your general comments can be summarized as 1) refocusing the research questions/objectives, 2) reorganization and consolidation of the introductory sections, 3) enhancing the glacier process side.

To address these concerns we plan to significantly reorganize the introductory section, and reduce its length to improve clarity and provide less general background and focus more on the specific study questions. We will also revisit the stated objectives to ensure that they are fully addressed within the text. We will separate the results and discussion sections. Separating the discussion section will allow us to expand the glacier process side and tie our results into the existing literature more explicitly, this should improve the impact of the manuscript. In making these changes will will address the general

comments you have outlined above, additionally we will address the detailed comments as outlined below.

Detailed comments: Page 2, line 1: rise in average glacier terminus elevation?

Response: We will reword for clarity.

2,7: Huggel et al should be 2003

Response: The tagline on the top of the paper says: Proceedings of EARSeL-LISSIG-Workshop Observing our Cryosphere from Space, Bern, March 11 – 13, 2002. Can you please confirm if the date should be 2003, or are you referring to a different paper?

2, 19 ff.: I don't think too many people would agree on the distinction of four primary methods to quantify glacier change. A more classical distinction foresees a distinction of glaciological, geodetic and hydrological methods to determine glacier mass change. I think the four methods identified here are somewhat odd, or arbitrarily identified (e.g. I would not see energy balance modeling as a method to determine glacier mass change). This also relates to page 2, 29 which is an incorrect statement in my opinion.

Response: We will significantly edit the introductory sections consolidating much of this material. We will address your concerns within this edited version.

3,4: you may add that energy balance studies are typically limited to point measurements of a weather station.

Response: Agreed.

3, 24: 10-100 m rather than decimeter (and vertical accuracy may also be indicated with a range)

Response: This is an error, should be decameter not decimeter. However, we will edit accordingly and include a range in the interest of improved clarity.
5, 2: there are other recent studies using UAV in the tropical Andes but they are not yet published, at least to my knowledge.

Response: We are aware of groups attempting UAV work in the tropical Andes, and in the Cordillera Blanca, but have not yet seen any results from this work. We will edit to address this.

7, 3: spell out SfM the first time used

Response: Agreed, we will correct this. We spelled it out in 4.4 but missed the brief first mention of SfM in section 4.2.

7, 22: do you mean outdated for use. . .?

Response: Yes, we will edit to improve clarity.

10, 10: I suggest to compare measured elevation changes to other available measurements in the Cord. Blanca.

Response: We will include this in the expanded and separated discussion section.

11, 25 ff: Section 5.3 is very thin. How compare these velocities to other debris covered glaciers? What are the implications?

Response: We will include this in the expanded and separated discussion section.

13, 14: did the UAV fly over 5000m? In the method section it is mentioned that flight height was 100 m above ground which would translate into a max of ca. 4800 m asl. In case you have experiences flying over 5000 m it may be interesting to know and report (but be specific).

Response: We have flown at elevations over 5000m, including at the glacier above Cuchillacocha (Quilcayhuanca valley), and more recently at Huaytapallana in Huancayo. However, for Llaca we did not need to fly above 5000m. We will clarify this point, expand as necessary and report on our experiences elsewhere in the expanded

discussion.

13, 15: the insights into the processes that control glacier melt could be enhanced in the main body of the text (see above comment).

Response: Agreed. We will expand and improve the glacier process side of this paper in the revised manuscript.

List of references: please check. Some references are duplicated.

Response: Agreed.

Figure 7: how was the glacier boundary determined? This is often difficult on debris covered glaciers.

Response: The glacier boundary was determined through visual inspection of the ortho imagery and the DEM surface in planimetric and 3D views. We followed terrain breaks and textural differences. There is some error in doing this, however it is much easier to identify the boundary with the high resolution imagery and DEM than when using coarser satellite derived data.

It would be helpful if the location of figures 10, 11, 12 could be indicated

Response: We will include extent rectangles for these figures.

in Fig. 7. The color scale should be improved: especially the 0 value level could be made more clearly visible.

Response: We will improve the colour scale and consider using a categorical scale bar as suggested by the anonymous reviewer.

---

## Author Comment (AC3) · 8 Jul 2017

The authors would like to thank the anonymous reviewer who took the time to provide a detailed review of this manuscript. Their constructive comments and suggestions will greatly improve the quality and impact of the manuscript. Our responses can be found below a reiteration of the reviewers comment.

The paper presents UAV survey results from the Cordillera Blanca, Peru. The data are novel, in that they provide information on a comparatively little-studied area and add

[Figure]

to the limited number of UAV surveys on high-altitude glaciers. However, for me, the study does not really address its objectives. It measures changes over a portion of the glacier, but does not capture the full volume change. I also question why the velocities were determined manually and for 72 points, rather than using more spatially comprehensive feature tracking. For objective three, the study does not explicitly investigate the role of debris thickness and its links to melt: it identifies debris as a secondary control, because it is thick at the terminus, but does not look at smaller scale variations or assess its impact on melt rates at higher elevations. It also does not consider that the debris may well be the reason for the tongues existence and characteristics (e.g. presence of ice cliffs). Overall, the discussion of debris cover is a little simplistic. Similarly, I felt Objective 4 was not properly addressed, and was a more limited discussion of the expansion of melt pools between two surveys, with some speculation about the potential for rapid drainage and lake expansion. I'm not sure how much this section extends our understanding of the interaction between the glacier and the lake. Overall, the paper has some valuable and interesting data, but it needs to be more specific about what the data can actually tell us (in terms of glacier dynamics, ice loss, lake interactions etc.) and the objectives should be re-focused accordingly. It would be good to see more comparison of these processes (e.g. melt cliffs, pools, lake formation) with analogues in the Himalaya: we have quite a bit of data and process knowledge from there, so it would be could to link these results to this literature and to make the comparison between different regions and sizes of glacier.

Response: While we agree this study would be improved by surveying the entire glacier, this was unfortunately not possible given access, elevation and flight time constraints. We have elaborated on this in response to your specific comments below. We will specifically address these constraints and the limitations it places on our findings in the revised manuscript. We will revisit our list of objectives and ensure that they are appropriate for the data, and that they are comprehensively addressed. We will separate the results and discussion sections, and expand the discussion to incorporate a broader comparison of our results with the existing literature. To the extent that it is

beneficial we may include reference to Himalayan glaciers, given the larger volume of literature. However, we feel that direct comparison to this region is of limited benefit given the considerable differences in the hydrologic and climatic regimes of these two regions.

My detailed comments are below. Page 1 Line 10: What about thinning, as well as retreat?

Response: We will correct this statement to include both retreat and thinning.

Also, altering THE timing.

Response: Agreed.

Line 19: Why the only the tongue and not the whole glacier? This would be important for water resources.

Response: It would be great to survey the entire glacier, and we agree that to derive and accurate mass balance using this method it is imperative. However, the main accumulation zone is almost 3km horizontally and 1000-1500m higher (topping out at ∼6000m) in elevation than the calving face – and the nearest safe, accessible launch point. Covering these distances with our multirotor drone was not feasible given the serious limitations on total flight time at these elevations. Furthermore, to produce reliable DEM's using structure from motion accurate ground control targets/control points are required. Accessing these upper parts of the glacier is extremely difficult and was not possible for this study.

Line 23: I don't follow how this shows a continued connection to the glacier tongue: the pieces of detached ice could just be melting away in-situ, completely dynamically detached from the glacier.

Response: We will improve this statement. Basically, we measured low (1-4m/yr) velocities over the sediments below the glacier calving face. Suggesting that the glacier is still pushing these sediments down valley. This suggests at least some connection

with the glacier above. However, you are correct that this does not necessarily mean a direct continuation of ice within these sections. We agree that the changes observed here are likely due to pieces of ice melting away insitu within a sediment and ice matrix.

Line 27: First statement needs a reference and give % for the Cordillera Blanca.

Response: We will include this value.

Page 2 Line 1: Rise in glacier terminus elevation sounds odd.

Response: We will edit accordingly.

Line 8: is>are

Response: Agreed.

Line 13: what is meant by 'geomorphic change for a debris covered glacier'? Do you mean glacier change, e.g. shrinkage / retreat?

Response: We will more clearly define this statement.

Line 19: I'm not sure this paragraph is needed. It's pretty general. I think it would be better to just say which approach was used and why, maybe at the start of the methods. I feel like all of section 1.1. is stepping too far back in terms of explanation and would be much better if it cut out the general material about each method, and focused specifically on its previous application to the study area (or relevant comparison glaciers, e.g. debris-covered glaciers) or why it hasn't been used previously. The paper does link to the study area at points, but it would benefit from a tighter, more focused argument throughout.

Response: We agree with these statements and will consolidate the introductory sections of the manuscript accordingly. These issues were raised by the other reviewers and the editor and are discussed more fully in the general response.

Page 5 Line 4: I like that the objectives are clearly stated.

[Figure]

Line 15 & 16. Reference if possible.

Response: We can probably find a suitable reference for this, but it's fairly general at this point.

Line 26: How was this determined? By this study or previous work? Needs to be cited if it is the latter.

Response: This is from general knowledge and experience in the region, however there is likely a reference somewhere to support it. If not I can provide specific terminal elevations for a couple of nearby glaciers.

Page 7 Line 22: dated for use?

Response: We can improve the wording of this statement. Basically the SRTM data is too coarse and in some instances too old (dated) to provide accurate elevation data for planning autonomous flight paths close to the ground surface (i.e. under ∼40m). If we had access to a higher resolution and more recent DEM we could be more confident the drone would not hit anything and therefore fly it closer to the ground and increase the spatial resolution.

Line 28: Was this why only the lower portion of the glacier was surveyed?

Response: See above re flight time, distance and access constraints.

Page 8 Line 16: DEMs (remove the ')

Response: Agreed.

Page 9 Line 1: Why do this manually? Why not use e.g. Cosi-Corr or Imgraft? This should give you a much more spatially extensive velocity field, so you don't need to interpolate. The glacier is quite small, but 72 points does not seem a lot when you have a heterogenous surface (pools, cliffs, debris cover). Based on Line 5, I'm not clear if these offsets were then used to calculate the distance the cliffs were advected by the flow or to determine how much they had melted back (i.e. by taking out the

velocity component and then subtracting the two DEMs). If the data are being used to estimate ice cliff melt rates, then can we really accurately separate out the local flow velocity from the melt rates (which are likely to be a few tens of cm, to a couple of meters) using just 72 points?

Response: We initially attempted to use Cosi-Corr for this but experienced difficulty in obtaining a good result. This may be due to the significant changes occurring within the scene and a limited number of identified matching points by the algorithm. In the interests of time we decided it was best to proceed with a manual feature tracking approach. We arrived at 72 points after iteratively adding more points and determining only minor changes in the velocity field after ∼60 points were included. Obviously minor velocity variations are likely missed using this approach and a continuous velocity field would be preferable. We hope to explore the application of Cosi-Corr and Imgraft in our ongoing work in the area but do not plan to address this in this manuscript.

Clarification per line 5. We used the velocity vectors to shift (polynomial georectification) the 2015 orthomosaic to match the 2014 location, thereby removing the down valley motion that took place between the two dates. Measurements of ice cliff back wasting were then made manually by measuring offset distances between the two dates – i.e. we calculated how much they melted back between the two dates from the orthomosaics. There are obviously some errors involved in this method (likely much less than 1m), which could be reduced by better deriving the flow velocity (either with Cosi-Corr/Imgraft or by using more points). We agree that with ice cliff melt rates of tens of cms to a couple of meters this error could be particularly problematic. However, in this case measured ice cliff melt rates were 2-25m which is considerably greater than the potential error of using this method.

Page 10 Line 12: For me, this shows a limitation to addressing objective 2 (and 4, to a lesser extent). We can't assess total volume changes or its potential future evolution, without accounting for the entire glacier.

Response: Given the limitations discussed above it was not possible to survey the entire glacier. However, we agree that this does limit to some extent the ability to address objective 2. We plan to revisit the list of objectives and specifically address them – see general response.

Line 23: As noted above, how confident can we be of removing the horizontal velocities local to the ice cliff, when 72 points were used across the glacier?

Response: See above.

Line 27: Needs a reference.

Response: We will include one.

Line 31: Quite vague and speculative. Do you have any data on debris thickness?

Response: We do not have quantitative data on debris thickness, but from our visual observations it is extremely variable. In some areas it is well over 1m thick and in others less than 5cm. Generally debris thickness exceeds 50cm and is thus likely to provide and insulation for the ice beneath. Indeed insulation from this thick debris cover (along with a relatively large accumulation zone) is likely the reason this glacier extends to a much lower terminal elevations than others in the Cordillera Blanca.

Page 11 Line 6: I don't think you can say debris cover is secondary here. First, the differential melt associated with ice cliffs is at least partly due to the removal of the debris layer. Second, the presence of the thick debris layer may well be why this tongue still exists at these altitudes (you note earlier in the paper this is a comparatively low altitude for the region), so although it doesn't increase melt rates, it is still an important control on them. There are no in-situ measurements of melt versus debris thickness, so I think it is hard to make this statement, particularly as the study only focuses on the tongue where the debris cover is thick. Debris may well be important here for supressing melt, then accelerate melt further up.

Response: We agree with these statements. The role of variable debris thickness will

be more clearly discussed in the expanded (and separated) discussion section.

Line 10: I'm unclear how much of the change discussed earlier in 5.2. is due to this movement of large objects, versus net change. We need to distinguish the two for e.g. forecasting water resources.

Response: Correct, however without completing a survey of the entire glacier it is impossible to quantify this.

Line 32: How? It would be tricky for the lake water to get very far up glacier. Could also relate to basal topography.

Response: We will clarify this statement in the expanded discussion. The basal topography is unknown, however it is likely similar to other glaciers in the Cordillera Blanca. For these glaciers we have observed the rapid expansion of proglacial lakes as their glaciers have receded up valley. By looking at the local topography of Llaca and comparing to others in the area it appears there is about 600-900m of the Llaca glacier tongue that lies within the depression of its future lake bed. As Llaca continues to thin and retreat up valley the lake will expand bringing it in closer proximity to the steeper faces of the mountains above, which is likely to increase the risk of an outburst flood, as has been observed and studied elsewhere in the range.

Figure 4: I would make the dots bigger, so they are easier to see. A black outline would help.

Response: We can make these changes.

Figure 7: Make the scale bards categorised, rather than stretched, as it's easier to read of individual values.

Response: We agree categorized is easier to read values, however stretched provides a clearer indication of the spatial variability. We will see how it looks with categorized bars and change if appropriate/beneficial to data display.

Figures 9-12. It would be useful to have some context about where these sites are, e.g. using extentboxes on Fig. 1 or 2.

Response: Agreed, we will add these.

Finally, we would like to thank you again for your time and constructive comments on this manuscript.

---

## Author Comment (AC4) · 8 Jul 2017

The authors would like to thank the reviewers who took the time to provide constructive comments and critical insights on the manuscript. Addressing the issues raised and incorporating these suggestions will greatly improve the quality and impact of the manuscript.

The primary concerns raised by the three reviewers and summarized by Stokes (Editor) are an overly detailed background section and limited discussion of the results.

Additionally there were some concerns with the structure of the paper, specifically merging of the results and discussion sections. Two reviewers (anon. and Huggel) also questioned how well the defined objectives were addressed. Detailed comments were provided by all reviewers and the response to these is addressed individually under their respective reviewer threads.

Following the advice and recommendations of Stokes, Anon. and Huggel we will consolidate the introduction to improve clarity. We will revisit the objectives to ensure that these are appropriate for the scope of the paper and clearly addressed in the discussion section. Per Stokes' recommendation the results and discussion will be separated and expanded to address the objectives defined above. We will provide a clearer discussion of the results, with a particular focus on the glacier process side, as this was an issue raised by all three reviewers. Finally, specific comments and corrections will be addressed as discussed in the individual reviewer responses.

---

## Author Response (AR1)

The authors would like to thank the reviewers who took the time to provide constructive comments and critical insights on the manuscript. Addressing the issues raised and incorporating these suggestions will greatly improve the quality and impact of the manuscript.

The primary concerns raised by the three reviewers and summarized by Stokes (Editor) were an overly detailed background section and limited discussion of the results. Additionally there were some concerns with the structure of the paper, specifically merging of the results and discussion sections. Two reviewers (anon. and Huggel) also questioned how well the defined objectives were addressed. Detailed comments were provided by all reviewers and the response to these is addressed individually under their respective reviewer threads.

Following the advice and recommendations of the editor and reviewers we have consolidated the introduction to improve clarity, removing significant sections of background and condensing the remainder We have revisited the objectives to ensure that these are appropriate for the scope of the paper and clearly addressed in the discussion section. Per Stokes' recommendation the results and discussion have been separated. We have expanded the discussion section and included a greater focus on the glacier process side. We have also provided a discussion of the application of UAVs to glaciology. Finally, all specific comments and corrections have been addressed as listed in the individual reviewer comments and author response.

[revised manuscript text omitted]